# Tannery Wastewater Recalcitrant Compounds Foster the Selection of Fungi in Non-Sterile Conditions: A Pilot Scale Long-Term Test

**DOI:** 10.3390/ijerph18126348

**Published:** 2021-06-11

**Authors:** Francesco Spennati, Salvatore La China, Giovanna Siracusa, Simona Di Gregorio, Alessandra Bardi, Valeria Tigini, Gualtiero Mori, David Gabriel, Giulio Munz

**Affiliations:** 1Cer2co, Consorzio Cuoiodepur, 56020 Pisa, Italy; gualtiero.mori@cuoiodepur.it; 2Department of Life Sciences, University of Modena and Reggio-Emilia, 41125 Modena, Italy; salvatore.lachina@unimore.it; 3Department of Biology, University of Pisa, 56126 Pisa, Italy; giovanna.siracusa@biologia.unipi.it (G.S.); simona.digregorio@unipi.it (S.D.G.); 4Department of Civil and Environmental Engineering, University of Florence, 50139 Firenze, Italy; alessandrabardi85@gmail.com (A.B.); giulio.munz@unifi.it (G.M.); 5MUT, Department of Life Sciences and Systems Biology, University of Turin, 10125 Torino, Italy; vtigini@gmail.com; 6GENOCOV, Department of Chemical, Biological and Environmental Engineering, School of Engineering, Autonomous University of Barcelona, Bellaterra, 08193 Barcelona, Spain; david.gabriel@uab.cat

**Keywords:** biofilm, bioremediation, community structure, fungi, wastewater treatment

## Abstract

This study demonstrated that a microbial community dominated by fungi can be selected and maintained in the long-term under non-sterile conditions, in a pilot-scale packed-bed reactor fed with tannery wastewater. During the start-up phase, the reactor, filled with 0.6 m^3^ of polyurethane foam cubes, was inoculated with a pure culture of *Aspergillus tubingensis* and *Quebracho* tannin, a recalcitrant compound widely used by tannery industry, was used as sole carbon source in the feeding. During the start-up, fungi grew attached as biofilm in carriers that filled the packed-bed reactor. Subsequently, the reactor was tested for the removal of chemical oxygen demand (COD) from an exhaust tanning bath collected from tanneries. The entire experiment lasted 121 days and average removals of 29% and 23% of COD and dissolved organic carbon (DOC) from the tannins bath were achieved, respectively. The evolution of the microbial consortium (bacteria and fungi) was described through biomolecular analyses along the experiment and also developed as a function of the size of the support media.

## 1. Introduction

In recent years, new emerging organic compounds [1] such as endocrines disruptors [2], pharmaceuticals [3], plastics and other micropollutants [4] have become relevant in wastewater and water treatment. Since recalcitrant pollutants could remain in the discharged effluent [5], in recent years the concern about their impact on the population and ecosystem increased. Consequently, more stringent limits on the discharge of wastewater treatment plants (WWTP) were expected in the near future [6]. Conventional biological wastewater treatments are based on bacteria, which are able to tolerate diverse conditions, grow fast and efficiently degrade a rather broad range of pollutants. However, several compounds are recalcitrant for bacteria in activated sludge systems. For decades, fungi have been investigated for their potential in the removal of recalcitrant compounds. Fungal exoenzymes/enzymes are capable of breaking a wide range of strong chemical bonds and a combined ecosystem of fungi and bacteria might be synergistic, since fungi could break the chemical bonds of recalcitrant compounds, while bacteria could bring these compounds to a mineralization [7]. Nowadays, the potential of fungi in the removal of recalcitrant organic pollutants has been widely investigated. However, these organisms are seldom employed as bioremediation agents in wastewater. One of the main reasons could be attributed to the scarce knowledge concerning the optimal process to enhance fungal performance and due to their scarce competitiveness towards bacteria under non-sterile conditions [8]. Real wastewater autochthonous microorganisms could represent a stress source for inoculated fungi, decreasing their degradation potential [8,9]. The presence of bacteria in fungal-based systems can be particularly negative during the start-up, being able to damage the mycelium of inoculated fungi, to reduce or inhibit their metabolism [10] and to minimize/inhibit fungal enzymes activity [11]. As a consequence, bacteria are commonly outcompeted by fungi in the typical conditions of conventional WWTP [8,9]. Since fungi are natural degraders of tannins, previous studies were focused on the autochthonous fungal strains and on the removal of recalcitrant compounds from tannery effluents [12,13]. Tannins are polyphenolic compounds produced by plants and used for centuries in the vegetable tanning of leather. Tannins differ from other natural phenols in their ability to precipitate proteins [14], and they are used in the tanning process to bind to the collagen matrix of animal skin and made the leather more durable and resistant to decay. Tannins represent one of the less biodegradable substances in tannery wastewaters and a high concentration of tannins could additionally inhibit biological treatment [15]. Despite the antimicrobial properties of tannins, several fungi, bacteria and yeast are quite resistant to tannins and, eventually, able to use tannins as carbon and energy sources; however, in natural environment, fungi seem to own higher degradative abilities towards tannins compared to bacteria. Previous studies in lab-scale reactor demonstrated that in presence of suitable process conditions [16] and specifically designed reactors [17] a selected fungal strain, *Aspergillus tubingensis* (AT) MUT 990, could be maintained. That study was performed in non-sterile conditions, inoculating the selected fungal strain on polyurethane foam (PUF) cubic carriers and continuously feeding reactors with a condensed tannin (*Quebracho* tannin) (QT) [18]. However, fungi are not yet applied as affective environmental biotechnology in WWTP for the removal of recalcitrant compounds and proper process design and conditions still need to be investigated.

In this study, a fungal-based system was applied to treat recalcitrant effluents. The system was operated under not-sterile conditions for 121 days and process conditions allowed fungal biomass to attach to the carriers and grow as biofilm. As a first step, a synthetic wastewater prepared with QT was employed as feeding, while, after 66 days, it was replaced with real exhausted tannin bath collected from a tannery. According to our knowledge, this is the first study of a fungal pilot-scale bioreactor employed to treat real wastewater in a long-term operation under non-sterile conditions.

## 2. Materials and Methods

### 2.1. Inoculum Preparation

The fungal strain inoculated (AT) was isolated from commercial tannin powder and it was preserved at the Mycotheca Universitatis Taurinensis collection on malt extract agar (MEA) at 4 °C. This strain was used in previous experiments with lab-scale reactors [17]. AT was inoculated onto generic MEA plates (150 mm diameter) and incubated at 25 °C in the dark for seven days. After the incubation, collected spores of AT from 10 plates were homogenized with 200 mL of physiological solution (0.9% NaCl) under sterile conditions. Six L of GLucose and Yeast liquid medium (GLY) were prepared in six bottles (1 L effective volume and about 1.1 L total volume), which were autoclaved at 121 °C for 30 min. At the same time, two air stone diffusers and plastic pipettes and sixteen PUF cubes (2 cm per side) were autoclaved, while two glass jars with lid (total volume 5 L) were sterilized using UV lamp for 1 h. Each glass vessel was equipped in a UV-cabinet (Top Safe, Euroclone, Italy) as follows: 1 stone diffuser, 8 PUF cubes, 3 L of GLY solution and 100 mL of homogenized fungi and inlet and outlet air pipes were connected to 0.2 µm air filters. Finally, the reactors were connected with the air pumps and incubated at room temperature for 64 days. At the end of incubation, the GLY medium was removed from vessels under sterile conditions. Before the inoculation in the pilot reactor, the biomass was weighed, completely homogenized and poured into the reactor with 8 L fresh GLY solution as described in Section 2.3. The immobilization on the carriers of the reactor occurred during the start-up phase in non-sterile condition.

### 2.2. Experimental Set-Up

The pilot reactor was constructed by Italprogetti Spa (San Miniato, Italy) and installed at the Cuoiodepur tannery WWTP (San Miniato, Italy) where start-up and tests were performed. The design of the pilot-scale reactor was based on previous lab-scale tests results [17]. A schematic layout of the piping and a diagram of the instruments is shown in Figure 1. Providing an overview of the pilot reactor employed in the current study. The pilot reactor was a cylindrical polypropylene vessel with a volume of 1.44 m^3^ equipped with a 600 L rotating cage. The cage could rotate from 1 to 3 rpm and was composed by four removable sectors filled with carriers. Each sector of the cage was filled with carriers of different sizes to evaluate the optimal size for fungal immobilization. The first and the second sectors were filled each with 6000 PUF cubes of 2.5 cm (side length), the third one with 620 PUF cubes of 5 cm (side length), the last one with 130 PUF cubes of 7.5 cm (side length). From now on, PUF size will be referred as small, medium and big for 2.5, 5.0 and 7.5 cm, respectively. On the bottom of the reactor, an air diffuser was connected to an air pump (Medo compressor, Nitto Kohki CO LTD, Japan) and a peristaltic recirculation pump (500 series, Watson and Marlow, USA) to avoid sedimentation. The feeding was pumped continuously (500 series, Watson and Marlow, USA) from an external, completely mixed, 1 m^3^ tank into recirculation line. The pilot reactor was connected to 500 L unmixed tank with a diluted HCl solution (0.1 M during the start-up and 0.05 M during the treatment phase). Two membrane pumps allowed the dosing of both reagents inside the reactor. A temperature probe, pH probe (Hach pHD sc, Loveland, CO, USA), redox probe (Hach ORP sc, Loveland, CO, USA) and dissolved oxygen probe (Hach LDO sc, Loveland, CO, USA) were installed inside the vessel. Actuators and probes were connected to the control panel in order to record the data, control the flow, maintain the pH and T set points and report malfunctions. The control system could maintain the pH set point by dosing HCl and temperature set point thanks to hot water recirculation (70 °C) in the internal pipes of the jacketed vessel. The temperature control was a low bound control, while the pH control was a high bound control, both designed to provide suitable conditions for fungal-based biomass. Further details were provided in Appendix A and Appendix A.

### 2.3. Process Operation: Start-Up

The two main critical issues of the start-up were: (1) avoiding the out-competition of the selected fungal strain by bacteria and (2) promoting mycelium growth inside PUFs. Process operations, selected taking into account both issues, were as follows: pH set point 5 ± 0.2; temperature set point 25 ± 2 °C; hydraulic retention time (HRT) of 3.4 days; cage rotation at 3 rpm for 1 min every 30 min; recirculation flow (4 L min^−1^) activated from the bottom to the top part of the reactor; aeration with low-bound at a DO concentration of 7 mg O_2_ L^−1^. Along the whole start-up phase, every 48 h, the 1 m^3^ feeding tank (continuously mixed) was cleaned and renewed with 1 g L^−1^ QT and a mineral medium containing, 0.1 g L^−1^ N-NH_4_Cl, 0.01 g L−1 P-H_3_PO_4_, and 3 mL of commercial antifoam dissolved in tap water. In the inlet solution, COD and DOC concentrations were quantified as 1645 ± 339 mg L^−1^ and 408 ± 79 mg L^−1^, respectively. Finally, a homogenized inoculum of AT was equally distributed into each sector in day 0. Reagents were of analytical grade except QT, which was provided by Chimont Spa (San Miniato, Italy). The start-up lasted 55 days.

### 2.4. Process Operation: Treatment Phase

The purpose of the subsequent treatment phase (duration 66 days) was to test the performance of the fungal biomass in the degradation of a tannin-rich effluent collected from a tannery factory after the vegetable tanning phase (exhausted tanning bath). In this second phase, the same process conditions used during the start-up were maintained. The supernatant of the effluent (characterization detail in Appendix A of Appendix A) was diluted 40 times with tap water in order to obtain a COD concentration comparable to the start-up phase (COD 1579 ± 331 mg L^−1^). Phosphorus was added as nutrient (0.01 g L^−1^ P-H_3_PO_4_) while nitrogen was already not limiting for the growth (about 20 mg N-NH_4_^+^ L^−1^).

### 2.5. Analytical Methods and Statistical Analysis

A sample of the inoculum, i.e., a pure culture of AT, was subjected to elementary analysis (C, H, N and S) through combustion at 1200 °C and an elemental analyzer (CHNS Flash 2000, Thermo Fisher Scientific, Spain). During the start-up and treatment phase, the system was sampled in the outlet to measure the COD, soluble COD (sCOD), and DOC concentrations. These parameters were measured from samples (filtered at 0.45 µm) using a DOC, a TOC analyzer (TOC-L series, Shimazdu Analyser, Japan) and Hach cuvettes (LCK 114). The previous measurements were performed three times per week, while the following analyses were performed once a week. Phosphorus, sulfate, nitrite, nitrate and ammonium were measured with Hach cuvettes and a Hach spectrophotometer. The dry mass of small PUFs (2.5 cm size), the total suspended solids (TSS) and volatile suspended solids (VSS) were measured in triplicate according to the standards methods.

### 2.6. Metagenomic DNA Extraction

The total DNA was extracted from PUFs collected during the start-up and treatment phases for the analysis of bacterial and fungal biodiversity. Samples were grouped based on the phase of treatment (inoculum, start up, middle and end) and, in the final part of the experiment, based on PUF size (Pufp, Pufm, Pufg). The total DNA was extracted and purified using the FastDNA Spin kit for soil (MP Biomedicals), according to the manufacture’s instruction. The Qubit 3.0 spectrofluorimeter (Thermo-Scientific, Waltham, MA, USA) was used for the measuring of quantity of extracted DNA, following the instruction for high sensitivity assay. The quality of the DNA was checked by calculating the ratio between the absorbances of the sample at 260/280 nm and 260/230 nm using the Spectrostar Nano UV-Vis spectrophotometer (BMG Labtech). A total of 200 ng of total DNA was used for sequencing. Illumina tag screening was performed on the V4-V5 hypervariable regions of genes coding for the 16S rRNA (16S rDNA) (primers 515F and 907R) [19,20] and the gene coding for the fungal Internal Transcribed Spacer 1 region (ITS1) (primer: ITS5–1737F and ITS2–2043R) [21] library construction and sequencing were performed by Novogene Co. Ltd. (Beijing, China).

### 2.7. Metabarcoding of the Fungal and Bacterial Biodiversity

The bacterial and fungal community in the pilot-reactor were tested using Illumina Miseq (Novogene Co., Ltd., Beijing, China) sequencing of V4-V5 16S hypervariable regions and ITS1 region respectively. To analyze fungal and bacterial taxa distribution, sampling was performed at the start-up and at the end of treatment; in the last case, samples were collected for each PUFs size. Obtained sequences were analyzed using a combination of Qiime 1.9.1 [22] and R packages. Raw data were firstly assembled in to paired-end reads using fastq-join (https://expressionanalysis.github.io/ea-utils/) (accessed on 4 March 2019). The paired reads were quality filtered and reads with a phred score Q < 20 were discarded. The files containing paired reads were converted into Fasta files using Fastx toolkit (http://hannonlab.cshl.edu/fastx_toolkit/) (accessed on 4 March 2019) and the obtained fasta files were imported into Qiime. Chimera check was performed using vsearch [23] and the reference chimera checked database ‘Gold’ in the Broad Microbiome Utilities (http://microbiomeutil.sourceforge.net) (accessed on 4 March 2019) operational taxonomic unit (OTU) table was obtained using the Qiime script pick_open_reference_otu.py, method sortmerna and sumaclust. Silva non-reduntant SSU database (128 release) was used with a confidence threshold of 0.97. OTU table, taxonomy, metadata and tree files were imported in R for alpha and beta diversity analysis using Phyloseq package [24]. Data preprocessing was carried out filtering sequences less than 0.1% in abundance and present in less than 2/3 of replicates. For statistical analysis, the OTU table was log transformed as previously described [25] via the implemented normalize.py script in Qiime [26]. Alpha diversity indices were calculated using diversity function in Vegan 2.5.4 package and the estimated richness function in phyloseq. One-way ANOVA and a Tukey test were used to estimate the statistical differences of alpha diversity indices among metadata categories. Weighted Unifrac distant matrix was used to perform principal coordinates analysis (PCoA) applying permutational analyses of variance (PERMANOVA statistical test). A similarity percentage analysis (SIMPER) was performed using simper function in Vegan package, to explore the OTU that contribute to differences among metadata categories. Summarized taxa table at phylum and genus level (where possible) were reported to highlight the phylogenetic groups that contribute to dissimilarities. To estimate the fold-change at genus level among metadata categories, DESeq2 package was used [27]. The shared OTUs in the different reactor phases were estimated by using Venn diagram, a freely online tool available at http://bioinformatics.psb.ugent.be/webtools/Venn/ (accessed on 2 April 2019) [28]. Detailed steps for the analysis of core microbiota are given along with raw script https://github.com/microbiome/tutorials/blob/master/Core.md. (accessed on 6 April 2019).

## 3. Results

### 3.1. Start-Up Phase

The elementary composition analysis of the fungus AT allowed the quantification of C/N and C/H ratio, which were approximately 13 and 7. At the beginning of the start-up the fungal biomass inoculated allowed to achieve a concentration of approximately 20 mg VSS L^−1^ in the reactor. During the start-up, biomass growth was observed in all the PUFs: at the end of the start-up (Day 55) the average dry mass concentration was 0.078 ± 0.018 g of dry mass in each small cube. It was observed that the small PUFs were the most effective, in terms of dry mass/PUF volume ratio, for biomass growth. In fact, such cubes had the highest dry mass density. Medium and big PUFs showed 2% and 32% lower average dry mass density that that in small cubes, respectively. Visual observation showed that mycelium did not grow deeper than approximately 2 cm, which can be due to transport limitation phenomena. Indeed, it was hypothesized that oxygen diffusion in the biofilm could be the limiting factor that hinder biofilm growth in the inner part of larger carriers [29]. Therefore, the estimated dry mass (from dry mass measurement on PUF) at the end of the start-up (55 days) was roughly 1.5 kg of dry mass in total. The estimated doubling time of the biomass was 8.5 days. Both visual and microscope observations (Appendix A in Appendix A) revealed a prevalent biofilm development in the small and medium PUFs. In conclusion, small and medium PUFs seem to be more suitable, compared to the other size tested, to promote fungal adhesion. As shown in Figure 2 and Figure 3, during the start-up, an average COD removal efficiency (RE) of 31% and a DOC RE of 21% were achieved. The average ratio among COD/DOC in the inlet and in the outlet was 2.5 ± 0.35 and 3.1 ± 0.2, respectively. In the outlet the average ratio VSS/TSS was 0.8 ± 0.2 and the concentration of VSS was 30 ± 14 mg VSS L^−1^. The estimated organic loading rate (OLR) was 20.0 ± 0.1 mg COD L^−1^ h^−1^ and the organic removal rate (ORR) 6.3 ± 0.9 mg COD L^−1^ h^−1^. These results show higher process efficiency in comparison to a previous laboratory scale bioreactor (with the same COD inlet and process conditions) in which the OLR was 27.6 mg COD L^−1^ h^−1^ and the correspondent was ORR 5.3 mg COD L^−1^ h^−1^ [17]. To sum up, in a QT-based medium, the selected operating conditions (established during the previous lab-scale tests) were confirmed to promote fungal growth and adhesion in the pilot-scale reactor under non-sterile conditions without any co-substrate.

### 3.2. Treatment Phase

In the treatment phase, in which tanning bath was employed as feeding, average inlet COD was 1579 ± 331 mg L^−1^, while DOC was 396 ± 40 mg L^−1^. In this phase, average COD and DOC removals were 29 ± 4% and 23 ± 7%, respectively. The trends of COD and DOC during the whole experiment are shown in Figure 2 and Figure 3. Since inlet COD and DOC concentrations employed in the start-up and treatment phase were similar, also removal performances achieved in the two phases were similar. The ratio between COD and DOC in the inlet and in the outlet were 4.1 ± 1.1 and 3.6 ± 0.3, respectively. The estimated OLR was 19.2 mg COD L^−1^ h^−1^ and the ORR 5.9 ± 1.0 mg COD L^−1^ h^−1^, similar to the previous phase.

The dry mass inside small PUFs grew till 0.13 g for each PUF during the experimentation (day 110). In the outlet, 28 ± 14 mg of VSS L^−1^ were detected. Consequently, the dry mass of biofilm of the overall reactor was estimated about 1.5 kg at the end of start-up period and 2.8 kg at the end of treatment phase. At the end of the start-up, the biomass present in the reactor was estimated to be approximately to 2.2 kg of COD. A further increase of 1.8 kg was quantified during the treatment phase. A COD mass balance during the entire start-up revealed that approximately 11.8 kg of COD were removed and 12.7 kg of COD were removed during the treatment phase. The biomass in the outlet was 0.7 kg as VSS in the start-up phase (approximate as 1 kg of COD) and in the treatment phase was 0.8 kg as VSS (approximate as 1.1 kg of COD). Equations (1) and (2) were employed to calculate observed growth yield (Y_obs_) for start-up (Y_obs_1_) and treatment phase (Y_obs_2_), respectively. Y_obs_1_ was very similar to Y_obs_2_:Y_obs_1_ = (2.20 Kg COD + 0.98 Kg COD (Biomass))/(11.78 Kg COD (Substrate)) = 0.27 (Kg COD)/(Kg COD) (1)
Y_obs_2_ = (1.77 Kg COD + 1.10 Kg COD (Biomass))/(12.69 Kg COD (Substrate)) = 0.23 (Kg COD)/(Kg COD)(2)

When assuming the final value of 2.2 kg of dry mass and considering that the only biomass leaving the system was present in the effluent, the system was operated at very high sludge retention time (SRT) (equal to 184 d when estimated with Equation (3), even though steady state was not achieved). The operational SRT was, however, longer than needed when considering the estimated doubling time of the biomass (8.5 days). The effect of the adsorption of tannins on mycelium, was not taken into account. Although adsorption could play a role in the whole process, its estimation is not an easy step. However, considering the dry mass in the PUFs and the degraded COD (mg), it is clear that biodegradation is the main process causing COD removal.
SRT = (Average mass of biofilm)/(average rate of biofilm detachment)(3)

In brief, the pilot-scale reactor, during the treatment phase, was successful employed to treat a real effluent: a tannin bath with a high concentration of QT and other tannins collected from a tannery belonging to the district of Tannery in Tuscany.

### 3.3. Microbial Taxa Distribution Changes along the Process

Orders with relative abundance less than 1% were included in “NA” category. Considering the first part of treatment (sample inoculum, start up and middle) some differences occurred in terms of orders abundances among the different PUFs size. While orders like *Xanthomonodales*, *Sphingobacteriales*, *Rhodospirillales*, *Enterobacteriales*, *Bacteroidales* and the most abundant *Rhodobacteriales* showed neglectable changes (false discovery rated, FDR: *p* < 0.05), *Acidobacteriales*, *Burkholderiales*, *Stramenopiles*, *Chlamydiales* and *Rhizobiales* appeared to be differently distributed among the different phases. *Acidobacteriales* and *Burkholderiales* abundances increased at the end phase of the treatment (FDR: t = 12.62, *p* = 0.001; t = 7.18, *p* = 0.001; respectively). *Chlamydiales* and *Rhizobiales* were more abundant in the first phase (FDR: t = 2.04, *p* = 0.003; t = 2.51, *p* = 0.02; t = 2.16, *p* = 0.02; respectively). According to taxonomic annotation, 3 fungal phyla (*Ascomycota*, *Basidiomycota* and *Ochrophyta*), 8 classes, 12 orders, 18 families and 22 genera characterized the fungal community. Sampling for fungal community were performed at the beginning and at the end of the treatment. As showed in Figure 4B, the orders *Tremellales*, *Hypocreales* and *Coniochaetales* showed evident changes in terms of abundance along the treatment. After 121 days of treatment the fungal community was mostly represented by the following orders: *Eurotiales* (31.46%), *Ophiostomatales* (24.70%), *Tremellales* (13.97%), *Hypocreales* (12.85%), *Capnodiales* (7.08%), *Saccharomycetales* (6.71%) and *Coniochaetales* (1.71%). It is worth noting that the order *Eurotiales*, to which AT belongs, did not show relevant changes along the treatment although the higher relative abundance percentages could be observed in medium and big PUFs. Sampling for bacterial community analyses was performed at different steps of treatment: Inoculum phase, start-up, middle treatment and end of the treatment phase (considering each PUF size). According to OTU classification, after 121 days, the bacterial community in the bioreactor hosted 19 phyla, 48 classes, 76 orders, 96 families and 99 genera. At the order level (Figure 4A), the bioreactor bacterial community was dominated by *Rhodospirillales* (45.1%), followed by *Acidobacteriales* (26%), *Chlamidiales* (8.3%), *Rhizobiales* (6.5%), *Burkholderiales* (5.8%), *Xanthomonadales* (4.1%), *Stramenopiles* (2.1%).

### 3.4. Bacterial Diversity Indexes and Core Microbiome

Bacterial alpha diversity indexes calculated in the bioreactor are reported in Appendix A (Appendix A and Appendix A). One-way ANOVA and the Tukey-test were applied to analyze the differences. Simpson index showed statistically significant differences in the comparison between the start-up and the endB groups (end of treatment), indicating that during the treatment the microbial community composition increased in terms of number of species. A total of 68 OTUs were present in all phases of treatment, representing the core microbiome (51.12% of total OTU retrieved) (Figure 5A). Core microbiome size and OTUs were retrieved using microbiome R packages. As shown in the heatmap (Figure 5B), the core microbiome consisted of fifteen orders, differently distributed among samples. Considering the total of sample (prevalence 100%), the most abundant order were represented by *Clamidiales* (relative abundance 0.1%), followed by *Actinomycetales, Bacteroidales, Rhizobiales* and *Stramenopiles* (relative abundance 0.01%). Most of the discovered orders were detected in 70–80% of samples with a low relative abundance (green square). The distribution of detected order suggested a possible key role for OTU belonging to the *Chlamydiales* order, since their distribution among samples was over-riding feature, starting from the lowest detection threshold (0.01%), for which they were distributed in all samples (prevalence 100%), to the highest detection threshold (20%), for which they were distributed among 20% of the dataset. Principal coordinate analysis (PCoA) revealed a clustering of bacteria at the OTU level (97% identity) according to the different phases of the process (inoculum, start up, middle phases and end phases). The interaction of the process phases affected significantly the beta-diversity (PERMANOVA: df = 5, F = 1.9769, *p* = 0.01598), explaining 63.9% of the total observed variations. The remaining part (36.1%) of the observed variance remain unresolved and cannot be explained considering only these fac tors. To calculate the dissimilarities among phases, due to the different family contribution, SIMPER analysis was performed (Appendix A). The highest bacterial community dissimilarity (39%) was detected between the start-up and the end of process. *Xanthomonadaceae* (13.5%), *Rhabdochlamydiaceae* (8.01%), *Acidobacteriaceae* (4.83%) are the major families that contribute to the differentiation of the two phases. Changes of bacterial composition among the beginning and the end of the process were calculated using DESeq2 fold change. In Figure 6 are represented the OTUs at genus level that significantly changes during the process. An increase is reported for *Acidosoma*, *Sulfurospirillum*, *Burkholderia* and *Mycobacterium* spp. A total of 34 genera decreased in terms of abundance during the process. The genus with higher decrease in abundance was *Ochrobactrum* sp.

## 4. Discussion

### 4.1. Reactor Performance and Advanced Application

In the present study, a successful long-term operation (121 days) and efficient treatment performance of real effluent were achieved: QT as the main carbon source, and specific operating conditions, were able to promote fungal growth under non-sterile conditions. In literature, few fungal-based reactors, operated at lab-scale and/or pilot-scale, were reported as stable and effective under non-sterile conditions for wastewater treatment and recalcitrant removal [8,9]. At the moment, only three fungal pilot-scale reactors studies are available in literature; even though, some of them were able to reach interesting performance, none of them described a long-term treatment. The first one employed an agitated reactor (working volume 110 L) with a fungal consortium for treatment of diary wastewater for 32 days with co-substrate addition (2 g/L whey) [30]. The second one employed a membrane reactor (170 L) for a short-term treatment (15 days) of a synthetic textile wastewater using a mixed culture of *Aspergillus versicolor* and *Rhizopus arrhizus* [31]. The third one employed a pilot-scale trickle-bed reactor (working volume 30 L) inoculated with *Pleurotus ostreatus* for the degradation of endocrine disruptors; such reactor was installed in a wastewater treatment plant and operated with some stability problem for 10 days [32]. One of the novelties of this study was represented by the target recalcitrant compound (QT and tannins in general), that was chosen with the aim of overcoming the out-competition by bacteria in long-term operation and of allowing fungi to attach to the carriers and grow in non-sterile condition. Since fungi are more resistant than bacteria to the inhibitory effect of tannins [33] QT allowed fungal biofilm development inside the reactor. The inhibitory mechanisms of QT and tannins in general could be further investigated, including the different mechanisms of action of bacteria versus fungi and the possible interference of bacteria quorum sensing. These investigations could further enlarge the exploitations of fungi in bioremediation. Interestingly, the first fungal quorum sensing molecule (QSM) discovered, Farnesol, acts also as a bacterial quorum quenching molecule (QQM) that significantly interferes with the production of the extracellular polymeric substances, reducing, for example, membrane biofouling [34]. In fact, tannins and in particular QT were known for their antifouling features [35]. Thus, it is possible that, in the treatment presented, QT could have acted as an efficacy QQM as one of its mechanisms of action against bacteria while fungi were intrinsically able to better cope with tannins. It is worth noting that the values of Yobs and RE did not show evident modifications between start-up (synthetic solution of QT) and treatment (real effluent). This similarity provides further evidence of the role of QT as crucial process parameter in the tested conditions. Even though, in the present study, the removal of COD was not complete, the RE was clearly higher compared to the one that would have been obtained with conventional processes based on bacteria. Hence, a scale-up of the fungal based technology can be evaluated in a tannery WWTP: in fact, vegetable tannin baths are widely mixed with the others tannery effluents with the consequence of significantly contributing to recalcitrant COD in the final effluent. Since the flow rate of highly concentrated recalcitrant COD of tanning bath is very small compared to overall effluent flow rate, a segregation of waste streams for treatment, as proposed in the present study, could be beneficial, even though rarely applied [36]. In the literature, other authors suggest vegetable tanning bath treatment based on nanofiltration, ultrafiltration [37], electrocoagulation process [38], treatment with ozone [39], precipitation and adsorption [40]. Even though these alternative treatments of the separated tanning bath are possible, the technology based on fungal biomass could provide advantages from environmental and economic point of view, including low operational costs; the acid tanning baths pH could reduce chemicals consumption to maintain pH setpoint. Furthermore, fungal-based systems could be employed in the co-treatment of other wastewater streams for the removal of their refractory fraction (i.e., tannery and pharmaceuticals) [41]. Moreover, the reactor design, proposed in this study, could be exploited as a technology for fungal biomass production aimed at different purposes, including bioaugmentation in activated sludge treatment [30], the clean-up of contaminated soil and sediments [42], or as a bio-flocculant (to improve settling) for treatments based on microalgae [43]. According to our knowledge, little is currently known on fungal yields and growth. However, the Y_obs_ calculated in this study suggest the need of taking into account the difference in fungal biomass growth compared to bacterial one when scaling-up fungal biomass production processes.

### 4.2. Structure Community Analysis

In the present study, the Eurotiales order and, in particular, the genus *Aspergillus* (Appendix A) resulted to be dominant in the start-up phase, confirming the success of attachment process of AT. The comparison of process performance and structure community analysis is limited by the few previous works available: as far as we know there were no other fungal-based reactor for tannins degradation, except the lab-scale test reported in Spennati [17]. The performance efficiency of the lab-scale test was similar to the present study; in both reactors there was a development a microbial community and *Aspergillus* was present during all the experiment, suggesting the capability of the strain to establish a synergism with other fungal species eventually deriving from the wastewater to be treated. An inoculum of AT resulted in a relative abundance of *Aspergillus* of 23% at the end of the experiment [17] close to the result achieved in the present test (approximately 30%). The inoculum could have played a crucial role in the composition of the microbial degradation consortium [44]. The process conditions have been shown to be a critical driver for consortium composition in long-term operations [45]; however, different inocula were able to produce distinctive microbial consortia with similar degradation capacities [46]. AT could have acted as promoter and accelerator for the formation of the biofilm: in fact, it was recently studied as catalyzer the formation of aerobic granules [47] and its use was investigated also in a combined system with microalgae [48]. Future investigations in similar long-term experiment will include a deeper investigation on the role of AT within the microbial community. Current literature demonstrated that the presence of other fungal species belonging to the *Ophiostomatales*, *Tremellales*, *Hypocreales*, *Capnodiales*, *Saccharomycetales*, and *Coniochaetales* orders could also play a role in tannins degradation. It is reasonable to assume that fungi belonging to the describer order these latter derive from the tannin solution used in the start-up phase. Their relative abundance became significant in the subsequent stages of the process (121 days). As showed in Figure 4B, the orders *Tremellales*, *Hypocreales* and *Coniochaetales* changed in terms of abundance along the treatment: in a study of Kazartsev et al. [49], these orders were described as part of wood-inhabiting fungal community, supporting the hypothesis of their resistance to tannin solution and suggesting an important metabolic role in the degradation process observed in the present study. It is important to highlight that PUF size did not appear to affect fungal community. *Saccharomycetales* are yeasts, and most of them did not develop a mycelium and did not colonized the PUFs, so their presence was probably only suspended in the water. The fungal inoculum was associated with the presence of a bacterial community, mainly characterized by bacteria from the order of *Rhizobiales*, *Rhodospirillales* followed by *Xanthomonadales*. *Rhodospirillales* were known to be present in different environmental niches such as water logged soils and puddy fields [50] where they were involved in sulfur oxidation. The bacteria metabolic degradation paths and their ecological niches might give some hints about their presence in the reactor: the *Rhizobiales* were generally present in the treatment of wastewater matrices by the formation of bacterial granules in full scale plant [51] or in reverse osmosis membrane reactor in biofouling [52]. *Rhizobiales* (*Alphaproteobacteria*) are well-known beneficial partners in plant-microbe interactions and recently described also as benefical partners of fungi [53]. Han et al. [54] assigned the genus *Denitratimonas* to *Xanthomonodales* order as denitrifying bacteria isolated from a bioreactor for tannery wastewater treatment. Molecular studies identified core microbial communities in efficient activated sludge WWTP and lack of these communities may point out to reasons for malfunctioning of wastewater-treatment system [55]. The bacterial community described for the inoculum changed during the start-up phase. Specifically, the bacterial community in the PUFs resulted to be composed by *Rhodospirillales*, *Chlamydiales* and *Rhizobiales*. The different ratio of sulfur/nitrogen could be a driver for the dominance of *Rhodospirillales*. While the increase of abundance of bacterial orders such as *Acidobacteria*, during the middle and start-up phases, was in accordance with the operational condition (pH 5) that was maintained in order to favor fungal growth toward bacteria [56]. This community structure description here reported will support the literature research aimed in the clarification of the relation of fungal and bacteria in environmental bioreactors [57] and provides a first step for future investigation about the relationship between the fungal inoculum and its influence on the long-term equilibrium of the microbial ecosystem reactors.

## 5. Conclusions

The experiment demonstrated that it is possible: (1) to grow fungal biomass under non-sterile conditions and (2) to partially remove tannins from a tannin-rich wastewater in the conditions tested. Moreover, the reactor design and process proposed could be exploited as a technology to produce fungal biomass for other purposes related to recalcitrant compounds removal. Fungi are effective in the biodegradation and/or depolymerization of most compounds recalcitrant to bacterial consortia, but the lack of knowledge hinder their application as effective environmental biotechnology. We believe that the findings obtained represent the first step for a future real-scale application the development of more effective biological processes, based on integrated use of fungi and bacteria, would allow to improve effluent quality and treatment sustainability. Moreover, the results suggests that further studies on tannins inhibitions and QQM mechanism versus bacteria could enlarge the exploitations of fungi in bioremediation.

## Figures and Tables

**Figure 1 ijerph-18-06348-f001:**
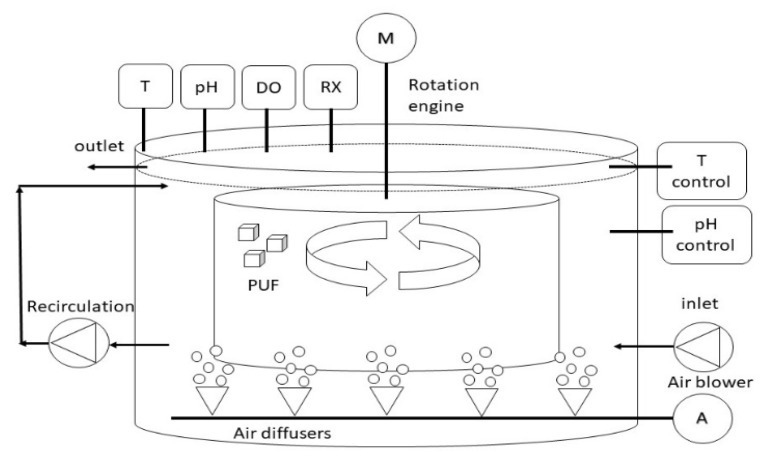
Schematic layout of the pilot-scale reactor. Temperature probe (T), dissolved oxygen probe (DO), redox probe (RX), polyurethane foam carriers (PUF), engine for rotation (M).

**Figure 2 ijerph-18-06348-f002:**
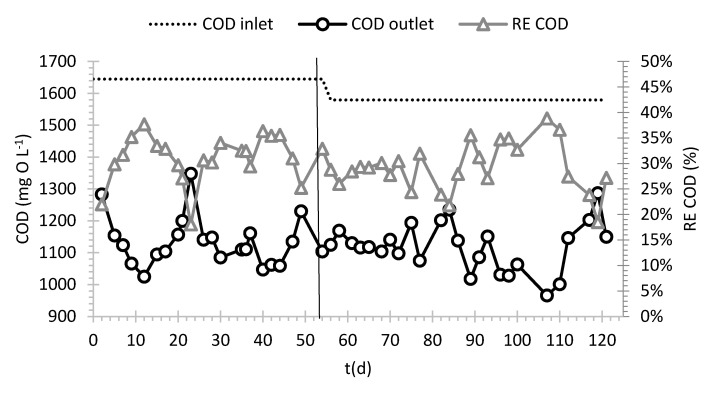
Inlet and outlet COD concentrations and COD removal percentages during continuous treatment in the pilot reactor. Different phases of the experiment are divided by vertical line.

**Figure 3 ijerph-18-06348-f003:**
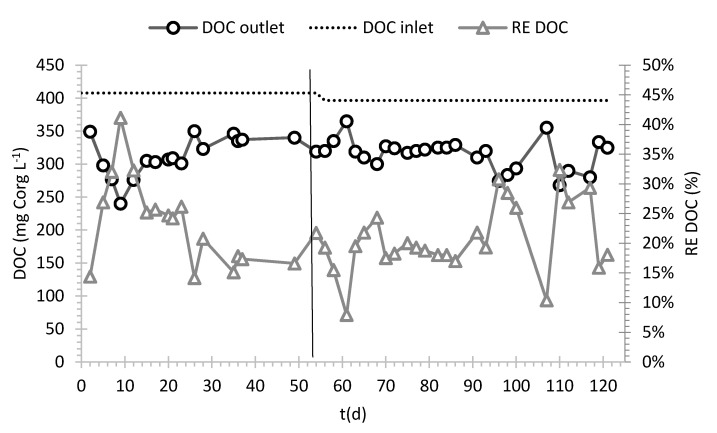
Inlet and outlet DOC and DOC removal percentages during continuous treatment in the pilot reactor. Different phases of the experiment are divided by vertical line.

**Figure 4 ijerph-18-06348-f004:**
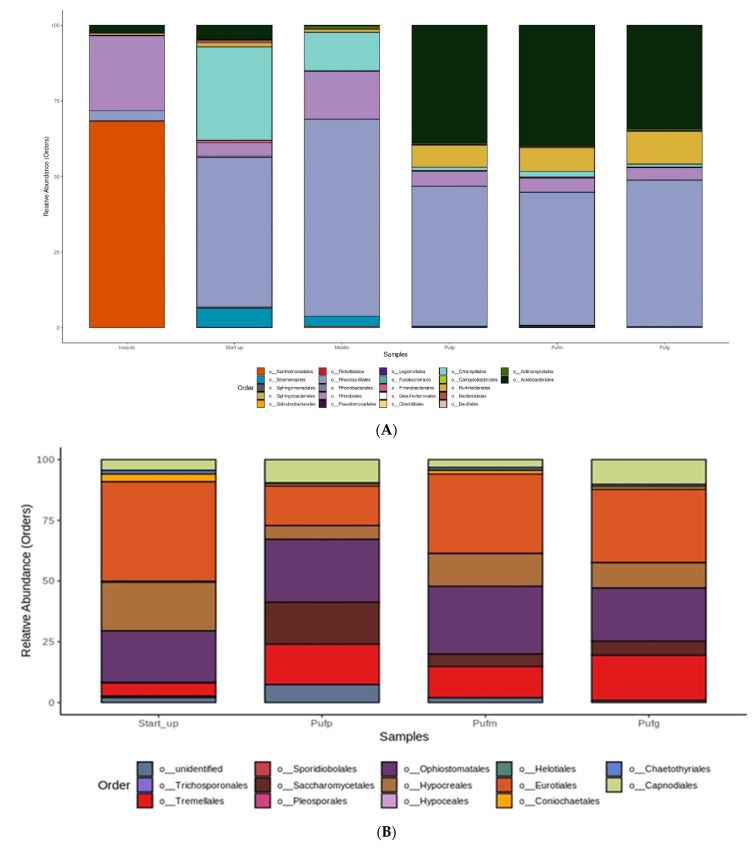
Relative abundance of different bacteria (**A**) and fungi (**B**) orders in the bioreactor. Out of all reads performed, in Figure 4 are shown those OTUs with more than 1% relative abundance and found in at least two thirds of replicates. Orders representing less than 1% of the total reads are grouped in ‘NA’. “Inocculum” t = 0; “start-up”: at the end of start-up phase t = 55 days; the “middle” t = 79 days; the “Pufp”: small PUFs, “Pufm”: medium PUFs and “Pufg”: big PUFs in the end of the treatment t = 121 days.

**Figure 5 ijerph-18-06348-f005:**
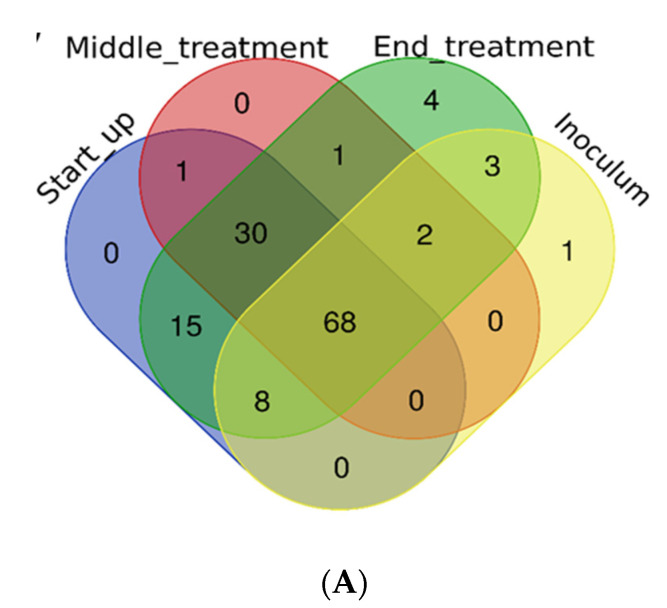
Comparison of microbial communities in samples at different phases of the process. (**A**) Venn diagram reporting the shared OTUs among phases. (**B**) Core microbiome heatmap at the families level. (**C**) Principal coordinates analysis (PCoA) for inoculum, start up, middle and end phases.

**Figure 6 ijerph-18-06348-f006:**
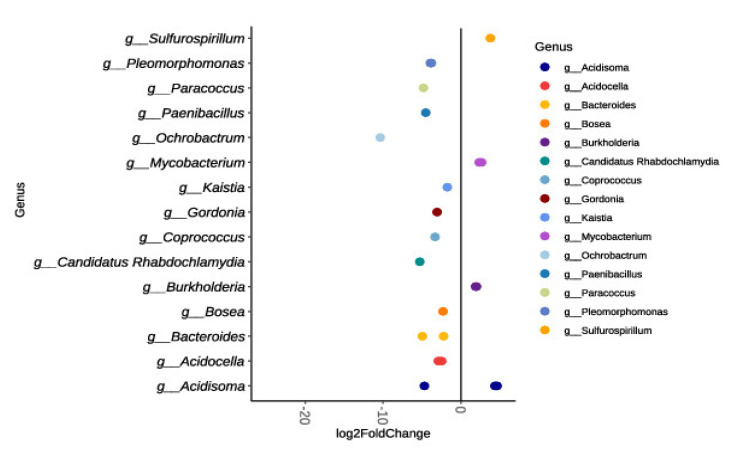
Log2 fold change among the first part (until the middle phase) and the end of the process in average normalized OTU counts grouped at the genus level.

## Data Availability

Not applicable.

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
