# Peer review of "Tannery Wastewater Recalcitrant Compounds Foster the Selection of Fungi in Non-Sterile Conditions: A Pilot Scale Long-Term Test"

_ijerph, 2021, doi:10.3390/ijerph18126348_

Round 1

Reviewer 1 Report

Just a question, why you started using synthetic wastewater prepared with QT  as feeding, instead to immedialy start with the real exhausted tannin bath collected from a tannery?

Author Response

Response to Reviewer 1 Comments

Point 1: Just a question, why you started using synthetic wastewater prepared with QT as feeding, instead to immedialy start with the real exhausted tannin bath collected from a tannery?

Response 1: Nowadays, a fungal-based bioreactor able to maintain stable fungal growth and performance, under sterile and non-sterile conditions, is still challenging. In previous experiments (Spennati et al., 2020, 2019), we were able to develop a fungal and bacterial consortia stable in the long-term operation of bioreactors (in non-sterile conditions) fed with QT after initial inoculation with Aspergillus tubingensis (immobilized in carriers). Reactor fed with QT outlasted the colonisation of bacteria and allowed a stable fungal biofilm able to remove QT. Fungi have been shown to be more resistant to the inhibitory effect of QT than bacteria, and the QT concentration therefore allowed for the maintenance of a stable fungal biofilm in non-sterile conditions. In this work we decided to use the synthetic wastewater prepared with QT for the feeding in the first part of our experiment in order to verified that the chosen conditions allow to promote fungal growth and immobilisation in the pilot-scale reactor in non-sterile (without pre-immobilization on carriers of the inoculum as we have done in lab-scale test). In the second part we demonstrate that the obtained biomass was able to growth and degraded a real exhausted tannin bath collected from a tannery. Thanks to those results in future experiments we could start with real exhausted tannin bath collected from a tannery.

Reference:

Spennati, F., Mora, M., Tigini, V., La China, S., Di Gregorio, S., Gabriel, D., Munz, G., 2019. Removal of Quebracho and Tara tannins in fungal bioreactors: performance and biofilm stability analysis. J. Environ. Manage. 231, 137–145.

Spennati, F., Ricotti, A., Mori, G., Siracusa, G., Becarelli, S., Gregorio, S.D., Tigini, V., Varese, G.C., Munz, G., 2020. The role of cosubstrate and mixing on fungal biofilm efficiency in the removal of tannins. Environ. Technol. (United Kingdom) 41, 3515–3523.

Reviewer 2 Report

Nowadays, wastewater treatment is a key issue in modern society and is receiving great attention around the world. The article entitled "Tannery wastewater recalcitrant compounds foster the selection of fungi in non-sterile conditions: a pilot scale long-term test" is an interesting study that seeks to serve as a basis for future work that incorporate fungi with to treat recalcitrant compounds, in this case.

The document is generally clear and well written. The manuscript is prepared in good order with detailed data. On the other hand, the relevance of the review is well reflected.

Clear and concise introduction where the subject matter of the issue is presented.

Robust results and discussion with a wealth of references to support the explanations.

English language and style are fine but minor spell check required. English language should be checked throughout the manuscript. In some points of the manuscript, the process description and explanations could be clearer.

The only thing I could point out would be that in the Conclusions section, it would be convenient to give more relevance to the advantages that fungi present or compared to other types of microorganisms or compounds. All this further enhances the importance of using fungi in these applications, shows the large number of advantages they present and that make them attractive to be able to follow this path of research.

For all these reasons, I RECOMMEND THEIR PUBLICATION in Journal of Environmental Research and Public Health in its present form.

Author Response

Response to Reviewer 2 Comments

Point 1: Nowadays, wastewater treatment is a key issue in modern society and is receiving great attention around the world. The article entitled "Tannery wastewater recalcitrant compounds foster the selection of fungi in non-sterile conditions: a pilot scale long-term test" is an interesting study that seeks to serve as a basis for future work that incorporate fungi with to treat recalcitrant compounds, in this case.

The document is generally clear and well written. The manuscript is prepared in good order with detailed data. On the other hand, the relevance of the review is well reflected.

Clear and concise introduction where the subject matter of the issue is presented.

Robust results and discussion with a wealth of references to support the explanations.

English language and style are fine but minor spell check required. English language should be checked throughout the manuscript. In some points of the manuscript, the process description and explanations could be clearer.

The only thing I could point out would be that in the Conclusions section, it would be convenient to give more relevance to the advantages that fungi present or compared to other types of microorganisms or compounds. All this further enhances the importance of using fungi in these applications, shows the large number of advantages they present and that make them attractive to be able to follow this path of research.

For all these reasons, I RECOMMEND THEIR PUBLICATION in Journal of Environmental Research and Public Health in its present form.

Response 1: We really thanks reviewer for the appreciation of our work, the accurate revision of the paper and the suggestions. We changed the conclusion of the manuscript according to the received request; moreover, the style has been carefully checked and many typos and errors have been corrected.

Reviewer 3 Report

The paper provides the description of bacterial and fungal consortia colonizing a pilot scale reactor treating tannery wastewater. The introduction fully describes the state of the art and the innovation of the study. Materials and methods needs to be implemented and few modifications in results section could help the reader.  As summarized above, there are few points that must be carefully addressed prior to publication and are depicted in the specific comments section:

Line 178-190: please provide more information regarding sequencing protocol

Line 183: please insert “was used” after “(Thermo-Scientific, USA)”

Line 189: please add the reference for primer pair

Line 193: replace “Myseq” with MiSeq”

Line 337-339: The patterns and colours of some orders in the bar are not easy to differentiate. May consider clarify.

Line 354-356: how does the core microbiome size was calculated? what is 11000 referring to? what is the unit of measurement?

Line 473: replace “systemwith” with “system with”

Line477: check the sentence after the colon

Please check along the entire text the use of the right reference style

Author Response

Response to Reviewer 3 Comments

Point 1: The paper provides the description of bacterial and fungal consortia colonizing a pilot scale reactor treating tannery wastewater. The introduction fully describes the state of the art and the innovation of the study. Materials and methods needs to be implemented and few modifications in results section could help the reader.  As summarized above, there are few points that must be carefully addressed prior to publication and are depicted in the specific comments section:

Line 178-190: please provide more information regarding sequencing protocol

Response 1: we are grateful to the Reviewer for the careful revision of our manuscript. We are sure that the comments and suggestions which were provided will surely improve the

quality of our article. In the revised version of the manuscript we address the criticisms which were pointed out by the Reviewer and the manuscript was accordingly modified. We improved the writing also corrected the reference format. we added further details and descriptions. We apologize for the hasty and rough writing of the submitted version. We hope that several aspects of this part have been now improved. We acknowledge that the sequencing protocol was not clearly explained, unfortunately, the sequencing was performed by Novogene company, making the protocol description not available. The sentence “Library construction and sequencing were performed by Novogene Co. Ltd. (Beijing, China)” was added to the main text.

Point 2: Line 183: please insert “was used” after “(Thermo-Scientific, USA)”

Response 2: Thank you for your suggestion. The correction was made.

Point 3: Line 189: please add the reference for primer pair

Response 3: we added the missing reference and we apologies for the mistake.

Point 4: Line 193: replace “Myseq” with MiSeq”

Response 4: Thank you for your suggestion. The correction was made.

Point 5: Line 337-339: The patterns and colours of some orders in the bar are not easy to differentiate. May consider clarify.

Response 5: Thank you for your suggestion. The color patterns were completely redefined, using more contrasting colors.

Point 6: Line 354-356: how does the core microbiome size was calculated? what is 11000 referring to? what is the unit of measurement?

Response 6: Thank you for your revision. The figure was redraw adding the x axys label in the right way. Also the legend was redefined, making the figure more clear. Part of the main target main text were completely reformulated

Point 7: Line 473: replace “systemwith” with “system with”

Response 7: Thank you for your suggestion. The correction was made.

Point 8: Line477: check the sentence after the colon

Response 8: The sentence was corrected, we apologies for the lack of clarity.

Point 9: Please check along the entire text the use of the right reference style

Response 9: Reference style has been carefully checked and many typos and errors

have been corrected in the reference list